# Psychological Distress, Resilience, and Help-Seeking Experiences of LGBTIQA+ People in Rural Australia

**DOI:** 10.3390/ijerph20042842

**Published:** 2023-02-06

**Authors:** Tamara Reynish, Ha Hoang, Heather Bridgman, Bróna Nic Giolla Easpaig

**Affiliations:** 1Centre for Rural Health, School of Health Sciences, College of Health and Medicine, University of Tasmania, Locked Bag 1322, Launceston, TAS 7250, Australia; 2College of Nursing and Midwifery, Charles Darwin University, Casuarina, NT 0810, Australia

**Keywords:** psychological distress, resilience, LGBTIQA+, rural, mental health

## Abstract

The aims of this paper were to explore mental health, the aspects associated with psychological distress and resilience, and the help-seeking experiences of LGBTIQA+ people. This research used a mixed-method approach using a survey and semi-structured interviews. The study was conducted in rural and remote Tasmania, Australia. Sixty-six participants completed the survey, and 30 participated in interviews. Participants reported a range of mental health concerns and varied experiences of accessing care and support in rural Australia. Depression and anxiety were the most common among participants. Almost half of all participants had attempted suicide in their lifetime, and just over a fifth had self-harmed. Two-thirds of the sample had high/very high psychological distress. For respondents, lacking social support was associated with increased psychological distress and low level of resilience. Public acceptance and social support enhanced interviewees’ resilience. Interviewees experienced a lack of nearby mental health professionals, were unimpeded by operating hours, and trusted mental health professionals, which impacted help seeking and mental health. The findings indicate that acceptance, access and proximity to care, and mental health professionals’ cultural competence would benefit rural Tasmanian LGBTIQA+ peoples’ mental health. There is a need to improve public education, improve mental health professionals’ curricula, and provide inclusive and tailored mental health care.

## 1. Introduction

Lesbian, gay, bisexual, transgender, intersex, queer/questioning, asexual, and people with additional sexual and gender identities, including pansexual, polyamorous, gender nonconforming, and many more (LGBTIQA+), have been found by other researchers to have higher mental illness rates than heterosexual, cisgender people [1]. This mental health disparity is due to discrimination, victimisation, exclusion, and other manifestations of minority stress [2,3]. Disparities can exacerbate psychological distress and impact resilience [4]. Resilience is the ability to bounce back from trauma, adversity, or considerable stress [5]. Resilience is not fixed, however, but dynamic and interactive, and can change overtime both in spite of and due to adversity [6].

### Literature Review

Protective factors cited elsewhere associated with resilience include social support, community connection, heterosexuality, and higher income [4]. These protective factors affect resilience via their many positive psychological (i.e., optimism and problem solving), biological (sympathetic nervous system), and social (i.e., attachment figures and role models) correlates [7]. Homosexuality, younger age, a lack of social support, and victimisation have been linked to psychological distress [4,8]. Rurality can also impact LGBTIQA+ peoples’ mental health as it tends to coincide with increased homophobia, isolation, identity concealment, and limited community belonging [9,10].

Researchers have found that LGBTIQA+ people access mental healthcare more than heterosexual people and from a range of mental health professionals (MHP) [11,12]. The usage difference is related to disparate mental health needs [13]. Due to specialised needs, such as gender-affirming care, mental health needs increase with increased intersecting identities [13]. Rurality has been found to impact uptake of mental healthcare by LGBTIQA+ people due to limited service access (including access to gender-affirming care), workforce shortages, and MHP lacking cultural competency [9,10]. Researchers have also found that rural LGBTIQA+ people are more likely to have higher levels of general psychological distress, suicide rates, and self-harm prevalence than those in metropolitan regions [3].

The island state of Tasmania, Australia, has characteristics of rurality. Tasmania has the highest socio-economic disadvantage, highest unemployment, and the lowest national education attainment rate [14]. The state has the second highest national suicide rate yet the fewest MHP [14,15]. While general practitioners (GP) are the most commonly consulted health professional for mental health issues, only 34% of Tasmanian GPs consider themselves adequately trained to provide psychological support [14,16].

This study, therefore, aimed to explore the mental health, the aspects associated with psychological distress and resilience, and the help-seeking experiences of LGBTIQA+ people in rural Tasmania, which can help improve MHP knowledge on minority populations. It was hypothesised that (1) transgender participants would encounter the most barriers to care; (2) that acceptance would have a positive direct link to resilience; (3) support would have a positive direct link to resilience; and (4) that absent social support would have a negative direct link to resilience.

## 2. Materials and Methods

The study design was based on queer theory and sexual configurations theory grounded in the human right of bodily autonomy. Queer theory explores the ways in which heteronormativity and cisnormativity are oppressive, emphasises the fluidities of genders and sexualities, and challenges related dualistic binaries [17,18]. Sexual configurations theory integrates the diversity of gender, sex, and sexuality in research and clinical practice [19]. Bodily autonomy is the right to self-determination for everyone, regardless of their sex, sexuality, and gender and without discrimination [20], making it a suitable basis for research into LGBTIQA+ people. Lived experience of mental health and help seeking were explored via a mixed-methods research approach [21]. Although, a mixed-methods approach is time consuming, challenging, and requires expertise in multiple areas, it was used due to many advantages. This approach produced a result that was more comprehensive than either qualitative or quantitative alone. Furthermore, the qualitative data validated, explained, expanded upon, or enhanced the quantitative data and vice versa to produce beneficial results while coalescing their diversity [21].

Data in this study represent a subset of a larger project which explored the mental health and related service use of LGBTIQA+ people, sex workers, and kink-oriented people in rural Tasmania. The larger project included a survey (*N* = 78) and interviews (*N* = 33). The current study is focused solely on LGBTIQA+ participants and includes a sample of 66 online survey respondents and 30 interview participants. Data from sex workers and kink-oriented participants have been reported separately.

### 2.1. Survey

The principal author’s clinical practice and previous findings informed survey development [22,23,24,25]. Six members of the public who fit the inclusion criteria were involved in the study development and design. These six piloted an early draft of the online survey to determine suitability and comprehension of questions and instructions, to uniform understanding of the questions, to aid in determining gaps in survey questions, and to ensure optimal functionality. They provided extensive feedback, which was incorporated. All reported completing a survey and five participated in interviews. Those who self-selected to receive results will be emailed publications arising from their participation.

The survey contained 174 questions that were organised based on identity (e.g., LGBTIQA+) and research topic (demographics, mental health, and service-use experiences). Demographics such as gender, sexual orientation, and relationship status were collected via multiple-choice survey questions and an open-ended ‘other’ option. The survey also contained questions on psychosocial variables (i.e., social support, disclosure, and victimisation) and help-seeking experiences (i.e., uptake, barriers, and facilitators), which were assessed via Likert scales and/or yes/no questions. The survey also contained the Kessler Psychological Distress Scale (K10) and Brief Resilience Scale (BRS).

The K10 is a 10-question assessment that uses a five-point Likert scale measure of mental distress experienced in the last 4 weeks [26]. Questions included ‘How often did you feel hopeless?’ and ‘How often did you feel worthless?’ Each question is scored from 1 for none of the time to 5 for all of the time. Scores for the ten questions were totalled, producing a score from 10 to 50. K10 scores were grouped into four levels of psychological distress: low (10–15), moderate (16–21), high (22–29), and very high (30–50) [27]. The K10 had excellent internal consistency, reliability, and validity (Cronbach’s alpha = 0.93) [26]. Cronbach’s alpha for the current study was 0.92.

The BRS is a 6-item assessment of a person’s ability to recover from stress with three positively worded statements (e.g., ‘I tend to bounce back quickly after hard times’) and three negatively worded ones (e.g., ‘I have a hard time making it through stressful events.’) to minimise response bias [5]. Likert scale response options ranged from 1 = strongly disagree to 5 = strongly agree. Totalled scores are mean scores of all answers ranging from 1 to 5. The higher the mean score is, then the greater the respondent’s resilience is. Mean scores indicated if participants had low (1.00–2.99), normal (3.00–4.30), or high (4.31–5.00) resilience (Smith et al., 2008). The BRS has good internal consistency and reliability (Cronbach’s alpha = 0.80–0.91) [5]. Cronbach’s alpha from this study’s sample was 0.93. The BRS has positive optimism (*r* = 0.69) and active coping (*r* = 0.38) correlations and negative anxiety (*r* = −0.53), depression (*r* = −0.50) and perceived stress (*r* = −0.61) correlations, demonstrating convergent and divergent validity [5].

### 2.2. Interviews

The primary author conducted two interviews with each interviewee held either in person or via telephone: a telephone-screening interview and a main interview. The screening interview consisted of four questions based on a protocol developed by Burke Drauker, Martsolf, and Poole [28]. The purpose of the screening interview was to preclude current acute psychological distress (which was achieved for all participants). The main interviews ranged between 15 and 60 min. Participants were asked a mix of 24 open- and closed-ended questions on mental health, risk and protective factors, and help seeking. Interviewees provided their demographic information in response to seven questions, including ‘What is your sexual orientation?’, ‘What is your gender?’, and ‘What pronouns do you use?’ Sample interview questions included, ‘What hurts your mental health and what improves it?’ and ‘What has been your experience with help seeking for mental health issues in rural Tasmania?’

### 2.3. Recruitment

The primary author recruited participants via convenience sampling across professional and personal networks and via snowball sampling through Facebook and third-party organisations. Thirty organisations assisted with recruitment, including local, state-based, and national support and advocacy services; community groups; not-for-profits; and private and public mental health services. Participants of one research tool were invited to participate in the other. Survey participants could enter a draw for a $100 Australian Dollar (AUD) gift voucher and interview participants received a $30 AUD gift voucher. Inclusion criteria required that participants self-identified as LGBTIQA+, were 18 years of age or older, were residents of rural Tasmania within the last 2 years (excluding the capital city and its surrounds), had preexisting mental health issues in their lifetime and used related formal or informal supports, and were proficient in English. Informed consent was required for participation. Interview participant recruitment continued until data saturation was attained in the concurrent data analyses [29]. The Tasmanian Social Sciences Human Research Ethics Committee approved the study (Reference #: H0018041).

### 2.4. Data Analysis

Quantitative data were analysed with IBM SPSS Statistics. Univariate analyses were conducted on all variables to generate descriptive statistics and frequencies. Bivariate correlation analyses were conducted with all survey questions against all K10 (low, moderate, high, and very high) and BRS (low normal, and high) scores to determine those significantly associated with each measure and to test each of the four hypotheses. For example, to test hypotheses 2, 3 (that acceptance and support would have a positive direct link to resilience) the survey question ‘My involvement in a BDSM kink community had a positive impact on my mental health’ was run against the BRS scores as well as the K10 scores. The majority of response categories were too small to be considered statistically appropriate to include. Those of sufficient size for consideration are included herein. Bivariate correlations were then conducted using Spearman correlation coefficient to determine if the hypotheses were nonsignificant (*p* > 0.05) and should be removed. The demographic categorical variables were age, gender, sexual orientation, postal code, K10 and BRS scores, relationship status, educational attainment, and employment status.

Interviews were transcribed in NVivo Transcription. Transcripts were verified for accuracy. NVivo qualitative data software was used to manage the qualitative data and facilitated analysis. The identified areas of examination for quantitative analysis (i.e., psychological distress, resilience, and help-seeking experiences) formed a framework for and structured the qualitative analytical inquiry. Across the continuum of inductive- to deductive-oriented approaches, the analysis veered closer to the latter given the a priori definition of its focus, scope, and parameters [30,31].

The initial steps common within methods of thematic analysis were used to identify and describe the relevant content [32,33]. This involved engaging in line-by-line coding of units of text explicitly or implicitly relevant to our framework and reviewing and refining these initial codes [32,33]. Within each identified area, codes were collated, aggregated, and revised to develop a descriptive narrative summary of these data. Qualitative data analysis offered a means to deepen and extend our understanding in relation to the quantitative findings.

## 3. Results

### 3.1. Overall Results

This study included 69.8% cisgender people and 30.2% transgender people. Transgender herein is an umbrella term for participants who were brotherboy, demigirl, gender questioning, gender nonconforming, or transgender. Participants ranged in age from 18 to 78 (Mean = 36.3, standard deviation = 14.1) and reported many sexual orientations, with bisexual the most common (28.4%) (Table 1).

All participants presented with a total of 31 current or historical issues; depression (79.2%) and anxiety (74.0%) were the most common. Almost half (46.9%) of all participants had attempted suicide in their lifetime and 20.8% had self-harmed. All participants experienced barriers and facilitators to help seeking.

### 3.2. Survey Results

#### 3.2.1. Psychological Distress

Sixty-five survey participants completed the K10 and scored an average of 26.1 (standard deviation = 8.6), indicating high levels of psychological distress (as hypothesised). A score of 22 or more or high and very high (H/VH) scores were considered clinically significant (requiring treatment to reduce severity). More than one-third (34.8%) of participants had high, and 30.3% had very high levels of distress; 66.2% had H/VH levels.

#### 3.2.2. Mental Health

Comorbid mental health diagnoses were reported by 86.4% of survey participants. Of the 19 issues reported by survey participants, depression (84.8%) and anxiety (81.8%) were the most common. Almost half (48.5%) of survey participants reported lifetime prevalence of suicidality, with 6.1% reporting suicide attempts in the previous year.

#### 3.2.3. Resilience

Of the 65 survey participants who completed the BRS, 46.2% scored low resilience, 50.8% scored normal, and 3.1% scored high. The average BRS score was 2.96 (standard deviation = 1.02), indicating low resilience. Depression and anxiety are also negatively associated with resilience [5]. Of those 46.2% with low resilience, 80.0% had depression, and 83.3% had anxiety. Of participants with normal resilience (50.8%), 87.9% had depression, and 75.8% had anxiety. Bivariate correlation analysis demonstrated correlations between ordinal variables that represented community, acceptance, support, and normal resilience and confirmed hypotheses 2, 3 (that acceptance and support would have a positive direct link to resilience). Of the 17 people who responded to the survey question, ‘My involvement in a BDSM kink community had a positive impact on my mental health’, 15 agreed, and of that 15, 75.0% scored normal resilience (*r* = 0.620 where *p* < 0.01, 2-tailed, *p* = 0.000). Table 2 demonstrates those survey questions that were significantly associated with normal resilience. Findings are as hypothesised.

#### 3.2.4. Help-Seeking Experiences

More than half (51.5%) of survey participants have seen an MHP for issues relating to being LGBTIQA+. Of this 51.5%, 70.6% have H/VH psychological distress, and 52.9% have low resilience. Regarding formal support, survey participants consulted private psychologists (78.1%), counsellors (67.2%), and psychiatrists (40.6%). Furthermore, the majority (89.4%) of survey respondents reported seeing a general practitioner (GP) for general mental health support. Nearly 45% reported seeing a GP for LGBTIQA+-related mental health issues.

#### 3.2.5. Barriers and Facilitators

Data representing barriers and facilitators were explored to test hypotheses. As hypothesised, cisgender participants experienced fewer barriers than transgender participants; that is, 76.7% of cisgender participants had an MHP located nearby (only 23.3% of transgender participants reported having local MHP), and 35.7% of cisgender participants lacked public transport (whereby 64.3% of transgender participants lacked public transportation). Additionally, 84.8% of participants saw a MHP who did not indicate that their diversity was a ‘phase.’ Trusting MHPs was not an issue for 71.2% of survey participants, and 58.8% saw an MHP who did not need to be educated about sex or sexual, or gender diversity (being LGBTIQA+). (Table 3).

### 3.3. Interview Results

#### 3.3.1. Mental Health and Contributing Aspects

Depression (*n* = 24) and anxiety (*n* = 19) were interviewees’ most commonly reported diagnoses. Depression findings were comparable for cisgender (*n* = 17/23) and transgender (*n* = 5/7) interviewees; however, more transgender interviewees experienced anxiety than cisgender interviewees (*n* = 6/7 and *n* = 13/23). Ten of the 30 interview participants reported lifetime suicide attempts: *n* = 4/9 transgender and *n* = 6/9 cisgender. Additionally, *n* = 22/30 reported comorbid diagnoses, and three of the remainder (*n* = 8/30) did not report a current diagnosis, but all (*N* = 30) experienced situational issues or contributing aspects (e.g., grief, financial difficulties, discrimination, or stigma).

Interviewees both had and lacked social support and community belonging. As hypothesised, their presence conferred a protective benefit against psychological harm and contributed to participants’ wellbeing; their absence harmed interviewees mental health and resulted in isolation and loneliness.


*[I]t was many years before I met somebody else that identified similarly to me, which is really hard in terms of your … mental health because [when] there’s no one around like you, how do you work out that you don’t come from bloody Mars … ? … So that really impacts on your mental health and … suicidality…*


Despite diagnoses and stressors, interviewees possessed the ability to recover from stress and bounce back via resilience.

*Resilience.* Interviewees demonstrated resilience in many ways: they believed in their ability to cope, stayed connected with supports, talked about their issues, helped others (humans and animals), and sought positivity and affirmations. Public acceptance and social support enhanced interviewees’ resilience. More than half benefitted from being seen positively via, for example, pro-LGBTIQA+ media coverage, legislative recognition of sex and gender diversity (*Births, Deaths and Marriages Registration Act* amendment), and rainbow flags on businesses. Public acceptance was described as, ‘[When] … a human [is] being seen in the world—truly seen—… that’s a gift.’ Almost all interviewees had friends, partners, LGBTIQA+ community members, or family that offered love, aid, or advice.

#### 3.3.2. Help-Seeking Experiences

Twenty-five interviewees consulted an MHP in rural Tasmania; four of the remainder reported previously seeing MHP in other parts of the country. Only one interviewee reported never having consulted an MHP. All interviewees relied on informal support from friends and loved ones. Faith-based services received the most backlash; one-third of interviewees expressly indicated they would not use them. Twenty interviewees consulted a GP in rural Tasmania for mental health issues, seeking support, medication, or referral to an MHP; 14 had negative experiences ranging from the GP was ‘visibly uncomfortable dealing with mental health’ to ‘he basically labelled me as a slut.’

The high prevalence of formal service usage enabled participants to assess its calibre. Interviewees encountered dozens of aspects that informed their service use/disuse; only those related to survey findings are reported on here.

#### 3.3.3. Barriers and Facilitators

Interview and survey data findings coalesced regarding barriers and facilitators. Specifically, interviewees experienced a lack of nearby MHPs (*n* = 20), were unimpeded by operating hours (*n* = 23), and trusted MHPs (*n* = 17), which impacted help seeking and mental health. Ability to obtain help from a trusted professional at convenient times encouraged help seeking; however, having to travel for care and take time off work to do so impeded uptake.


*I feel just that living, living in rural areas, you either see one of the few people [MHP] who are close by, in which case your ability to get help is somewhat limited by the professional’s time. Or you go to a larger area where it’s limited by your ability to travel that distance [so access to services and care is compromised].*


Unmet needs represented absent, insufficient, or inadequate care, which increased risk:


*Outside of rural areas, I’ve found government services for mental health quite encouraging as a whole. But within them [in rural areas] … I’ve not really found much help … the harm, as always, comes from the interminable waiting.*



*… funds for mental health services are limited in [rural Tasmania]. And there is a lot of pressure on the people delivering these services to keep up with the demand… And sometimes that means [a person] misses out [on getting psychological care]. … that makes me a bit reluctant to go to these services for fear of not being able to get what I need…*


Interviewees offered recommendations to address these gaps. Regarding the lack of MHP, they recommended additional government funding for existing services and new, specialised ones. Rural services’ adoption of community support, outreach support, and peer-support to enhance existing offerings was also endorsed. The incorporation of lived experience within a service would add community expertise and knowledge, thereby enhancing competency and capacity for all involved.

Twenty-six interviewees recommended training for MHPs, which could offset gaps in knowledge and cultural competence. Suggested training topics included empathy, diversity, body positivity, trauma, polyamory, inclusivity, and inclusive language. Basic training on gender and sexuality was also recommended.

## 4. Discussion

This examination of LGBTIQA+ people with preexisting mental health issues in rural Australia revealed high rates of depression, anxiety, and psychological distress. Participants also possessed resilience and embraced protective factors. Together, these findings indicate that overall, our participants have poor mental health. Indeed, inclusion criteria required that participants had preexisting experiences with situational and/or clinical mental health issues. Of the 96 participants in this study, only three interviewees did not have a diagnosis (they experienced situational stress). Other researchers have found that LGBTIQA+ people have been found to have higher mental illness rates than heterosexual, cisgender people [1]. However, related aspects of the findings of this study were unexpected.

National studies have revealed higher distress in people with mental disorders [34]. National studies into rural LGBTIQA+ people report psychological distress ranges of 22.8% to 57.2% [1,35]. Notably, 66.2% of our participants had H/VH psychological distress. National LGBTIQA+ studies also reported diagnoses of depression that range from 43.9% to 60.5% [1,35]. Overall, 79.2% of our participants experienced depression. National anxiety rates for LGBTIQA+ people range from 47.2% to 52.0%; 74.0% of our participants had anxiety. Marginality and minority stress accompany rurality and predilections for distress, depression, and anxiety in LGBTIQA+ people are well known [2,10,36]. However, national mental health strategies for LGBTIQA+ people fail to be implemented adequately, particularly in rural areas. Perhaps the urban-centric focus of these strategies is contributing to their poor implementation.

While the majority percentages indicate that the included aspects could be interpreted as facilitators to help seeking, the inverse of these majorities represent barriers. Our transgender participants experienced the most barriers to care, supporting hypothesis 1. Indeed, other research has concluded that gender diverse people experience, a wide range of barriers to accessing quality healthcare [11] as well as more barriers than cisgender people [13,37]. Considering the high rates of depression, anxiety, and distress in our transgender participants, these barriers are compelling.

Barriers to care also included 41.2% of participants who had to educate MHPs on LGBTIQA+ matters. This is perhaps not surprising given that a recent review of research in Australia, the US, the UK, and Canada identified the need for education, training, and support for healthcare providers as critical for the delivery of quality and culturally appropriate care for rural LGBTIQ+ communities [10]. Education and training can play a vital role in enhancing the confidence as well as the knowledge of healthcare providers for working with LGBT community members [38,39]. A US study found that 48.8% of physical health providers received no formal education on transgender healthcare during their training—citing transphobia as the reason for this gap [40]. Similar Tasmanian findings are unknown; however, given the high percentage of our participants who had to educate rural MHPs, one possible explanation is that transgender curricula content is equally sparce and that transphobia is equally prevalent. This barrier places greater burden on a population already experiencing disparities.

Hypotheses 2, 3, and 4 were supported. Acceptance and support presented herein as direct social support from loved ones and community belonging; indirect support presented as a belief in increased public acceptance; and self-acceptance was evident in participants’ self-pride. The absence of support presented as lacking social support. While the majority of our survey participants (50.8%) scored normal resilience, 46.2% scored low. When viewed against the high prevalence of distress, depression, and anxiety in our participants, the resilience scores and the ameliorative impact of a majority normal-resilience score are important. Complex self-identity could provide explanation; that is, the identity and self-concept development inherent in diverse populations can buffer against stress and distress [2]. That more than 40.0% of participants were also kink-oriented could add to the complexity of the self-identity, thus adding to the buffering effect. The majority normal-resilience score could also hinge on acceptance, which is a cornerstone to wellbeing, a fundamental human need, and the metonymy of social support, public acceptance, community belonging, inclusion, and self-pride.

Previous research has confirmed the importance of acceptance to LGBTIQA+ mental health [3,13]. The rural context of this study emphasises the need for the cultivation of acceptance on micro, meso, and macro levels due to increased isolation, prejudice, exclusion, and the harm caused by a lack of acceptance [41,42]. These findings suggest that approaches to building resilience and reducing distress may need to become integral facets of rural care delivery. Furthermore, given how acceptance, support, and community connection enhance a person’s resilience, perhaps pathways for support outside of mental health services—as provided via peer support groups, for example—may be an option for exploration.

## 5. Conclusions

LGBTIQA+ Tasmanians in this study experience chronic exclusion and erasure from mental health curricula, research, and policy. The impacts of this erasure have dire but preventable psychological implications. The present study has revealed unique findings regarding ways in which acceptance, access and proximity to care, and MHP cultural competence are vital for rural LGBTIQA+ people. Addressing these findings could ease distress, increase resilience, encourage help seeking, and optimise support offerings. Suicidality and mental health status require further research based on specific identities and intersectionalities. The mental health of the specific sub-identities of the target populations could be examined independently and in a more in-depth manner to as to draw conclusions unique to each orientation/identity. Transgender people’s experiences with psychological assessments causing harm and specifics regarding the ignorance of MHP could also be explored. Finally, this study’s sample size would be considered a limitation if representation and generalisability were the purpose of the study.

## Figures and Tables

**Table 1 ijerph-20-02842-t001:** Participants’ demographic characteristics (*N* = 96).

Characteristic	Survey (*N* = 66) No. (%)	Interview (*N* = 30) No. (%)	Total (*N* = 96) No. (%)
Gender			
Cisgender woman	35 (53.0)	16 (53.3)	51 (53.1)
Cisgender man	9 (13.6)	7 (23.2)	16 (16.7)
Nonbinary	10 (15.2)	2 (6.7)	12 (12.5)
Trans woman	5 (7.6)	2 (6.7)	7 (7.3)
Trans man	6 (9.1)	---	6 (6.3)
Additional genders	1 (1.5)	3 (10.0)	3 (3.1)
Intersex status			
Yes	2 (3.0)	---	2 (2.1)
Unsure/Prefer not to say	1 (1.5)	1 (3.3)	2 (2.1)
Sexual orientation ^b^			
Bisexual	20 (30.8)	7 (23.3)	27 (28.4)
Lesbian	13 (20.0)	3 (10.0)	16 (16.8)
Gay	8 (12.3)	2 (6.7)	10 (10.5)
Pansexual	7 (10.8)	3 (10.0)	10 (10.5)
Queer	7 (10.8)	3 (10.0)	10 (10.5)
Asexual	6 (9.2)	1 (3.3)	7 (7.4)
Additional sexualities	4 (6.2)	11 (36.7)	15 (15.8)
Kink/BDSM			
Kink oriented	33 (50.0)	6 (20.0)	39 (40.6)
Kink curious	---	1 (3.3)	1 (1.0)
Educational attainment ^c^			
Not university educated	35 (53.0)	8 (47.0)	43 (51.8)
University educated	31 (47.0)	9 (53.0)	40 (48.2)
Employment status ^b^			
Employed	33 (50.0)	22 (75.9)	55 (57.9)
Not employed	33 (50.0)	7 (24.1)	40 (42.1)
Relationship status ^a^			
Single	34 (51.5)	9 (40.9)	43 (48.9)
In a relationship	28 (42.4)	7 (31.8)	35 (39.8)
Polyamorous	4 (6.1)	4 (18.2)	8 (9.1)
Open relationship	---	2 (9.1)	2 (2.3)
Sex work experience			
Current or former	4 (5.1)	6 (18.2)	10 (10.4)
Australian Statistical Geography Standard (ASGA) remoteness class ^a^			
Inner regional (RA2)	28 (48.3)	17 (56.7)	45 (51.1)
Outer regional (RA3)	28 (48.3)	12 (40.0)	40 (45.5)
Remote (RA4)	2 (3.4)	1 (3.3)	3 (3.4)

^a^ Question optional, *n* = 88; ^b^ Question optional, *n* = 95; ^c^ Question optional, *n* = 83.

**Table 2 ijerph-20-02842-t002:** Factors Significantly Associated with Normal Resilience Among LGBTIQA+ Survey Participants in Rural Tasmania ^.

Survey Question	*N*	*n*	Normal BRS Score (%)	*r*	*p*
I have good social support from a significant other, family, friends, community or peers	51	28	54.9	0.277 *	0.026
I am proud of being sex, sexually, or gender diverse	34	33	63.6	0.299 *	0.016
Sex, sexual, or gender diversity has become more accepted over the past 5 years	65	60	56.7	0.314 *	0.011
BDSM/kink improves my mental health	20	19	73.4	0.579 **	0.001
I am proud of my kink orientation	16	14	63.4	0.447 *	0.010
My involvement in a BDSM/kink community has a positive impact on my mental health	17	15	75.0	0.620 **	0.000

*N* Total number of respondents who answered this survey question. *n* Total number of respondents who answered this survey question in the affirmative. ^ No survey question responses were found to be significantly correlated with high resilience. * *p* < 0.05, 2-tailed ** *p* < 0.01, 2-tailed.

**Table 3 ijerph-20-02842-t003:** Gender and formal help-seeking variables from survey data (*n* = 66).

	Cis Woman (*n* = 35)	Cis Man (*n* = 9)	Trans Woman (*n* = 5)	Trans Man (*n* = 7)	Non- Binary (*n* = 10)	Total %
External aspects						
*MHP said diversity a ‘phase’*	3 (8.6)	1 (11.1)	1 (20.0)	1 (14.3)	4 (40.0)	15.2
Proximity						
*MHP too far from home*	5 (14.3)	1 (11.1)	0 (0.0)	4 (57.1)	4 (40.0)	21.2
Access						
*Lack of MHP in my town*	5 (14.3)	4 (44.4)	2 (40.0)	4 (57.1)	5 (50.0)	30.3
*Limited operating hours*	7 (20.0)	3 (33.3)	2 (40.0)	4 (57.1)	5 (50.0)	31.8
*Lack of public transport*	4 (11.4)	1 (11.1)	0 (0.0)	5 (71.4)	4 (40.0)	21.2
Interal aspect						
*No trust for MHP*	6 (17.1)	2 (22.2)	2 (40.0)	3 (42.9)	6 (60.0)	28.8

## Data Availability

Not applicable.

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
