# Peer review of "Psychological Distress, Resilience, and Help-Seeking Experiences of LGBTIQA+ People in Rural Australia"

_ijerph, 2023, doi:10.3390/ijerph20042842_

Round 1
Reviewer 1 Report
The topic is very interesting
I have a few suggestions:
1. In my opinion, the title is better without the method stated in it, so I advise deleting it from the title
2. You repeat too many times the word This study in the abstract. Use other words like paper, research to avoid repetition. Keep the abstract simple, no numbers and results to be detailed there. Present briefly the main aim of the paper, the method, the findings and the novelty of paper without giving the details.
3. Extend the Introduction into a Literature review section. In the introduction keep the general parts, introducing the reader into the topic, the challenges and so on and extend the references in a Literature review section
4. I highly appreciate Discussions which put the results in perspective, mentioning other studies too
Conclusions should be extended to refer to theoretical and practical implications of the research, limits regarding the sample or the experience of the interviewers or others... and also future research directions.
5. References should respect the guidelines of the journal. Add in the literature review section a few more references especially from 2020-2022. Also in the Discussion section.
Otherwise congratulations for the topic and great success!
Author Response
The topic is very interesting
I have a few suggestions:
Comment 1: In my opinion, the title is better without the method stated in it, so I advise deleting it from the title.
Response: Thank you for the suggestion. “A mixed methods study” has now been removed from the title.
Comment 2: You repeat too many times the word This study in the abstract. Use other words like paper, research to avoid repetition. Keep the abstract simple, no numbers and results to be detailed there. Present briefly the main aim of the paper, the method, the findings and the novelty of paper without giving the details.
Response: The abstract has been revised accordingly. The only figures retained are those noted to report the number survey respondents and the number of interviewees, as we considered alternatives, but in this instance feel that this is most concise and clear way to communicate this information. Having removed the other details suggested, we do not believe that this detracts from the readability of the abstract.
Comment 3: Extend the Introduction into a Literature review section. In the introduction keep the general parts, introducing the reader into the topic, the challenges and so on and extend the references in a Literature review section
Response: The subheading “Literature review” is now included on page 1 to indicate the beginning of this section. We have also now incorporated two recent, highly relevant systematic reviews into this section and elsewhere in the paper. These are:
- Scott, Matt and Cornelius-White, Jeffrey H. D. 2022. “Mental health and social support experiences of transgender and gender nonconforming adults in rural America: A meta-synthesis.” Journal of Gay & Lesbian Mental Health, doi: 10.1080/19359705.2022.2128136
- Nic Giolla Easpaig Bróna, Reynish Tamara D., Hoang Ha, Bridgman Heather, Corvinus-Jones Sharon L., and Auckland Stuart. 2020. “A systematic review of the health and health care of rural sexual and gender minorities in the UK, USA, Canada, Australia and New Zealand.” Rural and Remote Health. 22, 6999. doi.org/10.22605/RRH6999
- Comment 4: I highly appreciate Discussions which put the results in perspective, mentioning other studies too
Conclusions should be extended to refer to theoretical and practical implications of the research, limits regarding the sample or the experience of the interviewers or others... and also future research directions.
Response: We have added information as suggested in the conclusions to address your comments.
Comment 5: References should respect the guidelines of the journal. Add in the literature review section a few more references especially from 2020-2022. Also in the Discussion section.
Response: We have now followed the guidelines of the journal for references. We also have added in the literature review and discussion section more recent relevant references from 2020-2022.
Otherwise congratulations for the topic and great success!
Thank you!

Reviewer 2 Report
The research is important and I recommend accepting it for publication, but several things must be corrected.
1- First, a more up-to-date overview of the benefits in the studied areas is needed and to address the differences from place to place. Particularly try to engage with previous writings about medicine\queer. Such as, Race, K. (2009). Pleasure consuming medicine: The queer politics of drugs. Duke University Press.
Shapiro, S., & Powell, T. (2017). Medical intervention and LGBT people: A brief history. Trauma, Resilience, and health promotion in LGBT Patients, 15-23.
Moll, J., Krieger, P., Heron, S. L., Joyce, C., & Moreno‐Walton, L. (2019). Attitudes, behavior, and comfort of emergency medicine residents in caring for LGBT patients: what do we know?. AEM education and training, 3(2), 129-135.
Ufomata, E., Eckstrand, K. L., Hasley, P., Jeong, K., Rubio, D., & Spagnoletti, C. (2018). Comprehensive internal medicine residency curriculum on primary care of patients who identify as LGBT. lgbt Health, 5(6), 375-380.
Bintley, H., & Winning, J. (2021). Embracing Difference: Towards an Understanding of Queer Identities in Medicine. The Mental Health and Wellbeing of Healthcare Practitioners: Research and Practice, 28-40.
Nelson, J. L. (2014). Medicine and making sense of queer lives. Hastings Center Report, 44(s4), S12-S16.
2- Secondly, a theoretical infrastructure must be provided on spaces that have been excluded from them for benefits, especially in medical contexts and have actually been revealed as safe spaces. Since the category of "exclusion" is central in this study, the author should demonstrate the argument by engagement with another research.
For example, the way Jewish communities dealt with AIDS. For instance - Ben-Lulu, E. (2021). “Who will say Kaddish for me”? The American Reform Jewish response to HIV/AIDS. Journal of Modern Jewish Studies, 20(1), 70-94.
Or the intersection not with religious discourse but maybe education:
Grant, R., & Nash, M. (2019). Educating queer sexual citizens? A feminist exploration of bisexual and queer young women’s sex education in Tasmania, Australia. Sex Education, 19(3), 313-328.
3- Third, provide more explanation regarding your specific decision of usage mix-methods, please cover the advantages or disadvantages in this kind of research.
4- Forth, how does this specific micro case illuminate on global LGBTQ experiences? Try in your conclusions to elaborate the conjunction between micro-macro, global-local.
Author Response
The research is important and I recommend accepting it for publication, but several things must be corrected.
Comment 1: First, a more up-to-date overview of the benefits in the studied areas is needed and to address the differences from place to place. Particularly try to engage with previous writings about medicine\queer. Such as, Race, K. (2009). Pleasure consuming medicine: The queer politics of drugs. Duke University Press.
Shapiro, S., & Powell, T. (2017). Medical intervention and LGBT people: A brief history. Trauma, Resilience, and health promotion in LGBT Patients, 15-23.
Moll, J., Krieger, P., Heron, S. L., Joyce, C., & Moreno‐Walton, L. (2019). Attitudes, behavior, and comfort of emergency medicine residents in caring for LGBT patients: what do we know?. AEM education and training, 3(2), 129-135.
Ufomata, E., Eckstrand, K. L., Hasley, P., Jeong, K., Rubio, D., & Spagnoletti, C. (2018). Comprehensive internal medicine residency curriculum on primary care of patients who identify as LGBT. lgbt Health, 5(6), 375-380.
Bintley, H., & Winning, J. (2021). Embracing Difference: Towards an Understanding of Queer Identities in Medicine. The Mental Health and Wellbeing of Healthcare Practitioners: Research and Practice, 28-40.
Nelson, J. L. (2014). Medicine and making sense of queer lives. Hastings Center Report, 44(s4), S12-S16.
Response: Thank you for the suggestions. We have considered this literature and decided to incorporate some of this literature into the “Discussion” section of the paper.
Comment 2: Secondly, a theoretical infrastructure must be provided on spaces that have been excluded from them for benefits, especially in medical contexts and have actually been revealed as safe spaces. Since the category of "exclusion" is central in this study, the author should demonstrate the argument by engagement with another research.
For example, the way Jewish communities dealt with AIDS. For instance - Ben-Lulu, E. (2021). “Who will say Kaddish for me”? The American Reform Jewish response to HIV/AIDS. Journal of Modern Jewish Studies, 20(1), 70-94.
Or the intersection not with religious discourse but maybe education:
Grant, R., & Nash, M. (2019). Educating queer sexual citizens? A feminist exploration of bisexual and queer young women’s sex education in Tasmania, Australia. Sex Education, 19(3), 313-328.
Response: Thank you for your suggestion which we will consider for our future paper.
Comment 3: Third, provide more explanation regarding your specific decision of usage mix-methods, please cover the advantages or disadvantages in this kind of research.
Response: We have added the information on our justification of using mixed methods approach.
Comment 4: Forth, how does this specific micro case illuminate on global LGBTQ experiences? Try in your conclusions to elaborate the conjunction between micro-macro, global-local.
Response: We have added information about this in the conclusion to address your comment.